# Willingness to use mental health counseling in diverse groups of Asian Americans

**Ronna Bañada[1], Yuri Jang[1,2,3]\*, Lawrence A. Palinkas[4]**

1 Suzanne Dworak-Peck School of Social Work, University of Southern California, Los Angeles, Los Angeles, CA, United States of America, 2 Edward R. Roybal Institute on Aging, University of Southern California, Los Angeles, Los Angeles, CA, United States of America, 3 Department of Social Welfare, Ewha Womans University, Seoul, Republic of Korea, 4 Herbert Wertheim School of Public Health, University of California, San Diego, San Diego, CA, United States of America

\* yurij@usc.edu

**Data Availability Statement:** Data are available from the dryad database (https://doi.org/10.5061/dryad.cjsxksnf0).

**Funding:** The support for data collection was provided by the City of Austin's Asian American

## Abstract

Responding to the underutilization of mental health services in Asian American communities, we examined factors associated with their willingness to use mental health counseling. Applying Andersen's Behavioral Health Service Model, we focused on the role of mental health needs and prior use of mental health counseling in shaping the attitudes toward mental health counseling of diverse groups of Asian Americans. We conducted a secondary analysis of data from 2,609 Asian Americans aged 18 or older who participated in the Asian American Quality of Life (AAQoL) survey conducted in central Texas. Logistic regression models of willingness to use mental health counseling were tested with predisposing (age, gender, marital status, education, nativity, and English-speaking ability), mental health needs (mental distress and self-rated mental health) and enabling (prior use of mental health counseling) variables. About 67% of the overall sample indicated their willingness to use mental health counseling. Individuals who met the criteria for mental distress showed 17% lower odds of willingness to use mental health counseling. The enabling role of prior use of mental health counseling was supported; those who had used counseling were over three times more likely to be willing to use counseling in the future than their counterparts without such an experience. Given the inverse association between mental health needs and the willingness to use mental health counseling, further attention should be paid to improving Asian Americans' recognition of mental health symptoms and awareness of the benefit of mental health services. The enabling role of prior use of counseling also highlights the importance of increasing the exposure to mental health services for Asian Americans. In efforts to promote mental health literacy, reduce cultural stigma, and advocate for mental health service use, consideration of cultural and linguistic diversity within the Asian American population is imperative.

## Introduction

Asian Americans are the fastest-growing racial/ethnic group in the United States, growing 81% between 2000 to 2019, from about 10.5 million to 18.9 million [1, 2]. Growing at a rate

Quality of Life initiative (Contract No. 26-8275-39, PI–YJ, Ph.D.).

**Competing interests:** There are no potential competing interests for all authors. No financial disclosures were reported by the authors of this paper.

four times faster than that of the overall US population, Asian Americans are projected to number at 46 million by 2060 [1, 3]. One of the major public health concerns about this growing population is the underutilization of mental health counseling [4–9]. According to national reports on mental health service use by US adults with any mental health illness, Asian Americans were three times less likely to use services than their non-Hispanic white counterparts [10].

Due to the stigmatization of mental health issues, many Asian Americans tend to express their symptoms through somatic complaints and are reluctant to use mental health specialty care [11]. However, studies with Asian American users of mental health counseling provide promising results in depressive symptom reduction and satisfaction with treatments [12–14]. Such findings call attention to strategies to promote the use of mental health counseling among this population with unmet mental health needs. Given the heightened mental health vulnerability in Asian Americans caused by the COVID-19 pandemic and anti-Asian racism, it is imperative to examine the underlying factors that may facilitate or deter the willingness of diverse groups of Asian Americans to use mental health counseling. Considering the diversity among Asian Americans, it is also important to attend to the similarities and differences in the patterns of mental health help-seeking across ethnic groups [5, 15].

With the fundamental components of predisposing, need, and enabling factors, Andersen's behavioral health model of service use is a useful guide to contextualizing individuals' willingness to use mental health counseling [16]. Predisposing factors include individual-level and sociodemographic characteristics which offer a background context for service use; studies report that service use is more likely among those with higher socioeconomic status [17–20]. Immigration-related factors such as nativity and English proficiency have also been used to contextualize willingness to use mental health counseling [21, 22].

Mental health status, represented by a formal diagnosis or self-reported symptoms, is indicative of the need for mental health services [15, 23, 24]. However, many individuals with mental health needs remain untreated, and such a gap has been a public health concern [25]. One of the popular measures of mental health status is the Kessler-6 (K6) [26]. Developed to assess nonspecific psychological distress for the U.S. National Health Interview Survey, K6 has been widely implemented among diverse racial/ethnic and age groups [26]. Another indicator of mental health needs is self-rated mental health, an individual's subjective assessment of their own mental health status [27–29]. Studies report that self-reported mental health is a meaningful indicator of personal recognition of mental health symptoms and perceived needs for service use [30]. We considered mental distress and self-rated mental health as connected but independent indicators of mental health needs.

The enabling variable of interest in the present assessment is prior use of mental health counseling. We conceptualized that prior use of mental health counseling would increase the likelihood that a person would be willing to seek mental health counseling. The exposure to mental health counseling would make a person familiarized with the service and open to its use. The positive impact of prior use of mental health counseling on the attitudes toward mental health services and actual service utilization has been observed in studies with diverse populations [31–33]. However, other findings have been mixed, for example, a study using the National Latino and Asian American Study (NLAAS) did not find a significant relationship between recent mental health services use and willingness to seek professional mental health services among Chinese, Vietnamese, and Filipino Americans [9].

Based on the aforementioned review, the aim of this study was to examine how the willingness of diverse ethnic groups of Asian Americans to use mental health counseling would be associated with mental health needs (mental distress and self-rated mental health) and enabling (prior use of mental health counseling) variables. We hypothesized that both mental

health needs and prior use of mental health counseling would be positively associated with the willingness to use mental health counseling. Recognizing the ethnic diversity within Asian Americans and the underrepresentation of non-English speaking individuals in population-based studies, in the present study we considered diverse ethnic groups of Asian Americans and included those with English language barriers.

## Methods

### Data

This study utilized a quantitative research design. Data came from the Asian American Quality of Life (AAQoL) survey, designed to explore the social and health needs of the growing population of Asian Americans in Central Texas [34]. The survey was conducted with self-identified Asian Americans aged 18 and older between February 1st to December 20th of 2015, in Austin, Texas. The AAQoL data were collected under the approval of the Institutional Review Board of the University of Texas at Austin (FWA#00002030). Prior to the survey, all participants signed a written consent form. The AAQoL survey questionnaire was originally developed in English and then translated into the native languages of five major Asian groups (Chinese, Hindi, Gujarati, Korean, Vietnamese, and Tagalog). The translation of the survey addresses the limitations in the availability of Asian languages in national surveys. The surveys were administered using a paper-and-pencil format, and participants chose the language of the questionnaire based on their preferences. To reach out to diverse groups of Asian Americans, surveys were collected at a total of 76 sites including churches, temples, grocery stores, small group gatherings, and cultural events. The AAQoL survey primarily used a convenience sampling approach, and efforts were made to reflect the ethnic composition of the Asian American population in the area. The 10-page surveys were self-administered; however, trained research assistants who were bilingual and bicultural were present at each of the survey sites to assist with the administration of the questionnaires. After removing cases with more than 10% missing information, the final sample size was decreased from 2,614 to 2,609. It is notable that over 48% of the participants used non-English versions of the questionnaire. More information on the sample and survey procedures is available elsewhere [34]. It should be noted that the AAQoL data were collected in central Texas before the COVID-19 pandemic, and the current study addresses the general aspects of mental health service use that may not be specific to location and time.

### Measures

*Willingness to use mental health counseling*. Participants were asked, "If you feel depressed, would you be willing to use mental health counseling?" Responses were coded into a binary format (0 = *no*, 1 = *yes*).

   *Mental health need*. Two indicators of mental health status were used. One is the Kessler-6 (K6), which measures non-specific psychological distress [26]. The K6 is a unidimensional scale with six questions about the frequency of experiencing distressful feelings within the past 30 days. The scale includes questions such as "How often did you feel nervous?", "How often did you feel so depressed that nothing could cheer you up?", and "How often did you feel that everything was an effort?" Each item was measured on a 5-point Likert scale of 0 (none of the time) to 4 (all of the time). Total scores range from 0 to 24, and the scale presents a high level of internal consistency ($\alpha$ = .88). A score of 6 or greater has been used as a cutoff for mental distress, and we used the binary variable (0 = *no mental distress*, 1 = *mental distress*) in the present assessment [26].

The other indicator of mental health needs was self-rated mental health. Participants were asked to rate their mental health on a 5-point scale. Responses were coded into a binary variable (0 = *excellent/very good/good*, 1 = *fair/poor*), following the strategies used in previous studies [27–29].

*Prior use of mental health counseling*. Participants were asked, "Have you ever used mental health counseling?" Responses were coded into a binary format (0 = *no*, 1 = *yes*).

*Predisposing variables*. Age (0 = 18–39, 1 = 40–59, 2 = 60+), gender (0 = *male*, 1 = *female*), ethnicity (0 = Chinese, 1 = Asian Indian, 2 = Korean, 3 = Vietnamese, 4 = Filipino, and 5 = Other), marital status (0 = *married*, 1 = *not married*), education (0 = ≥ high school graduation, 1 = < high school graduation), and nativity (0 = *U.S.-born*, 1 = *foreign-born*). English speaking ability was rated on a 4-point scale (not at all/not well/well/very well), and we recoded this variable to be binary as categorizing those who spoke English less than "very well" as a group with limited English proficiency (0 = English proficient, 1 = limited English proficiency).

## Analytical strategy

First, we reviewed the descriptive characteristics of the overall sample and ethnic subgroups. Because all study variables were in a binary or categorical format, percentage was reported, and ethnic group differences were evaluated by using Chi-square analyses. We used Chinese as a reference group because they are the largest and most-studied ethnic group in Asian Americans [35]. Prior to the multivariate analyses, we calculated Spearman's correlations among the study variables to understand their underlying associations and ensure the absence of collinearity. Using the overall sample and each of the ethnic groups, we conducted a series of logistic regression models of willingness to use mental health counseling. The predictors included (1) predisposing variables [age, gender, marital status, education, nativity, and English-speaking ability], (2) mental health need [mental distress and self-rated mental health], and (3) the enabling variable [prior use of mental health services]. Ethnicity was considered a predisposing variable in the analysis of the overall sample. All analyses were conducted using Stata/IC 16.1 [36].

## Results

### Descriptive statistics of the sample

Sample characteristics are summarized in Table 1. Diverse Asian subgroups were represented in the study (*N* = 2,609), including 640 Chinese (24.5%), 574 Asian Indians (22%), 471 Koreans (18.1%), 513 Vietnamese (19.7%), 265 Filipinos (10.2%), and 146 other Asians (5.6%). The other Asian group included Nepalese, Pakistani, Cambodian, and Japanese. Approximately half of the participants were between the ages of 18 and 39 (48.3%) and were female (55.2%). Over 90% of the overall participants were foreign-born and over 60% had limited English proficiency. Over 44% of the sample met the criteria for mental distress. Regarding self-rated mental health, 8.6% of the sample fell into the category of fair or poor self-rated mental health. About 10% of the sample had prior use of mental health services. Over 67% of the sample reported a willingness to utilize mental health services in the future.

Table 1 also reflects ethnic group differences. Compared to Chinese participants, those who identified as Vietnamese had a markedly higher rate of mental distress (54.6% vs. 38.9%). Asian Indians, Koreans, Vietnamese, Filipinos, and those identifying as other Asian ethnicities had lower rates of fair or poor self-rated mental health (5.1%-9.2%) than Chinese participants (13.7%). Compared to Chinese participants, Asian Indians had a notably lower rate of prior mental health service utilization (6.4% vs. 12.5%). Additionally, the proportion of the Chinese

Table 1. Descriptive Characteristics of the Sample (N = 2,609).

| Variable | % | | | | | | |
|---|---|---|---|---|---|---|---|
| | Overall sample (N = 2,609) | Chinese (n = 640) | Asian Indian (n = 574) | Korean (n = 471) | Vietnamese (n = 513) | Filipino (n = 265) | Other Asian (n = 146) |
| **Predisposing variable** | | | | | | | |
| Age | | | | | | | |
| 18−39 | 48.3 | 47.0 | 68.6*** | 38.9*** | 39.0*** | 42.2*** | 47.9** |
| 40−59 | 31.2 | 27.7 | 14.3*** | 40.2*** | 38.4*** | 41.4*** | 39.7** |
| 60+ | 20.5 | 25.3 | 17.1*** | 20.9*** | 22.4*** | 16.3*** | 12.3** |
| Female | 55.2 | 57.0 | 39.9*** | 60.5 | 57.5 | 70.0*** | 54.8 |
| Not married | 33.4 | 36.3 | 25.2*** | 25.7*** | 41.7 | 40.3 | 36.6 |
| < high school graduation | 18.6 | 14.2 | 7.6*** | 20.3** | 36.3*** | 16.2 | 20.0 |
| Foreign-born | 90.8 | 90.1 | 96.7 | 93.2 | 89.1 | 83.0 | 83.6 |
| Limited English proficiency | 62.4 | 71.7 | 44.8*** | 79.2** | 72.9 | 35.0*** | 49.3*** |
| **Mental health need** | | | | | | | |
| Mental distress (K6 ≥ 6) | 44.2 | 38.9 | 41.0 | 43.8 | 54.6*** | 41.6 | 48.9* |
| Self-rated mental health (fair or poor) | 8.6 | 13.7 | 5.1*** | 9.2* | 7.2*** | 6.5** | 7.5* |
| **Enabling variable** | | | | | | | |
| Prior use of mental health counseling | 9.7 | 12.5 | 6.4*** | 10.0 | 9.0 | 11.9 | 7.5 |
| **Outcome variable** | | | | | | | |
| Willingness to use mental health counseling | 67.1 | 77.8 | 57.8*** | 59.2*** | 74.7 | 65.1*** | 59.3*** |

Note. χ² analyses were conducted by comparing each ethnic group with Chinese

\* $p < .05$

\*\* $p < .01$

\*\*\* $p < .001$

sample with future willingness to use mental health counseling (77.8%) was higher than those of Asian Indians, Koreans, Filipinos, and those identifying with other Asian ethnicities (59.2%-65.1%).

## Predictors of the willingness to use mental health counseling

In bivariate correlations, there were no concerns about collinearity. The highest correlation was observed between nativity and English-speaking ability in the overall sample (Spearman's rho = .35, $p < .001$); those who were foreign-born were more likely to have limited English proficiency.

Findings from the logistic regression models of willingness to use mental health counseling are summarized in Table 2. In the overall sample, female participants had higher odds of future willingness to use mental health counseling than male participants. All ethnic groups, except Vietnamese, demonstrated significantly lower odds of future willingness compared to Chinese. Those with limited English proficiency also had lower odds of willingness to use counseling than their English-proficient counterparts. Regarding mental health status, those meeting the criteria for mental distress had 17% lower odds of future willingness to use counseling. No significance was observed in self-rated mental health. Prior use of mental health counseling was associated with 3.46 times higher odds of being willing to use mental health counseling.

Table 2 also summarizes the regression model in each Asian American subgroup. In all groups, female gender and prior use of mental health counseling were consistently associated with significantly higher odds of being willing to use mental health counseling in the future. While there were no age group differences in the overall sample, subgroup analysis showed

**Table 2. Regression model of willingness to use mental health counseling.**

| | Odds Ratio (95% Confidence Interval) | | | | | | |
|---|---|---|---|---|---|---|---|
| | **Overall Sample** | **Chinese** | **Asian Indian** | **Korean** | **Vietnamese** | **Filipino** | **Other Asian** |
| Age | | | | | | | |
| 18–39 | [reference] | | | | | | |
| 40–59 | 1.13 (0.91, 1.40) | 1.24 (1.00, 1.52)* | 1.11 (0.90, 1.37) | 1.27 (1.03, 1.56)* | 1.18 (0.96, 1.45) | 1.27 (1.03, 1.56)* | 1.26 (1.02, 1.54)* |
| 60+ | 1.15 (0.86, 1.53) | 1.14 (0.86, 1.52) | 1.12 (0.85, 1.49) | 1.19 (0.90, 1.58) | 1.18 (0.89, 1.57) | 1.19 (0.90, 1.57) | 1.17 (0.88, 1.55) |
| Female | 1.42 (1.19, 1.70)*** | 1.42 (1.19, 1.69)*** | 1.36 (1.14, 1.62)** | 1.45 (1.22, 1.72)*** | 1.43 (1.20, 1.70)*** | 1.45 (1.22, 1.73)*** | 1.43 (1.20, 1.70)*** |
| Ethnicity | | | | | | | |
| Chinese | [reference] | ----- | ----- | ----- | ----- | ----- | ----- |
| Asian Indian | 0.40 (0.30, 0.53)*** | ----- | ----- | ----- | ----- | ----- | ----- |
| Korean | 0.41 (0.31, 0.55)*** | ----- | ----- | ----- | ----- | ----- | ----- |
| Vietnamese | 0.87 (0.65, 1.18) | ----- | ----- | ----- | ----- | ----- | ----- |
| Filipino | 0.44 (0.31, 0.62)*** | ----- | ----- | ----- | ----- | ----- | ----- |
| Other | 0.39 (0.26, 0.57)*** | ----- | ----- | ----- | ----- | ----- | ----- |
| Not married | 1.10 (0.89, 1.36) | 1.20 (0.98, 1.48) | 1.17 (0.95, 1.44) | 1.23 (1.00, 1.51)* | 1.20 (0.98, 1.47) | 1.26 (1.03, 1.55)* | 1.25 (1.02, 1.54)* |
| < high school graduation | 0.91 (0.71, 1.16) | 1.02 (0.81, 1.30) | 0.91 (0.72, 1.16) | 0.93 (0.73, 1.18) | 0.85 (0.67, 1.09) | 0.94 (0.74, 1.19) | 0.94 (0.75, 1.19) |
| Foreign-born | 0.89 (0.62, 1.28) | 0.82 (0.58, 1.18) | 0.90 (0.63, 1.29) | 0.76 (0.53, 1.09) | 0.82 (0.58, 1.17) | 0.78 (0.55, 1.11) | 0.77 (0.65, 1.10) |
| Limited English proficiency | 0.72 (0.58, 0.89)** | 0.77 (0.63, 0.95)* | 0.77 (0.63, 0.95)* | 0.91 (0.74, 1.11) | 0.82 (0.67, 1.00) | 0.82 (0.67, 1.00) | 0.84 (0.69, 1.02) |
| Mental distress (K6 $\geq$ 6) | 0.83 (0.69, 0.99)* | 0.86 (0.72, 1.04) | 0.82 (0.68, 0.98)* | 0.82 (0.68, 0.98)* | 0.79 (0.66, 0.95)* | 0.82 (0.68, 0.98)* | 0.82 (0.69, 0.99)* |
| Self-rated mental health (fair or poor) | 0.84 (0.60, 1.17) | 0.80 (0.57, 1012) | 0.89 (0.64, 1.24) | 0.89 (0.64, 1.24) | 0.95 (0.68, 1.32) | 0.89 (0.64, 1.24) | 0.90 (0.65, 1.25) |
| Prior use of mental health counseling | 3.46 (2.32, 5.16)*** | 3.41 (2.30, 5.07)*** | 3.44 (2.32, 5.10)*** | 3.55 (2.40, 5.27)*** | 3.56 (2.40, 5.27)*** | 3.51 (2.37, 5.20)*** | 3.48 (2.35, 5.15)*** |

* p < .05

** p < .01

*** p < .001

higher odds of willingness to use mental health counseling among middle-aged (40–59) participants than those aged 18–39 in Chinese, Korean, Filipino, and other Asians. In Korean, Filipino, and other Asian groups, unmarried status was associated with a greater willingness to use mental health counseling. Limited English proficiency was significantly associated with lower odds of willingness in Chinese and Asian Indians. Mental distress was significantly associated with lower odds of willingness in all ethnic groups but Chinese.

## Discussion

To better understand the underutilization of mental health services in Asian Americans, we explored factors associated with their willingness to use mental health counseling. We paid particular attention to the role of mental health needs and prior use of services. The Asian American Quality of Life (AAQoL) survey offered unique data from diverse groups of Asian Americans including many individuals who were foreign-born and limited in English. The

data helped us reflect the experiences of diverse groups of Asian Americans with mental health and service use.

Our sample of Asian Americans revealed a high level of mental health concerns, with the overall prevalence rate of mental distress (44%) being markedly higher than that reported in a national sample of the general population in the U.S. (21%) [37]. In the present sample, Vietnamese participants were found to have the highest rate of mental distress at almost 55% while Chinese participants, at 39%, had the lowest rate. There was a wide range of ethnic differences in mental health needs and prior use of mental health counseling. For example, Asian Indians in the sample had a noticeably lower rate of prior mental health utilization of 6.4%. Over three-quarters of Chinese participants noted willingness to use mental health counseling, the proportion of which is considerably higher than Asian Indian, Korean, Filipino, and other Asian groups.

In the multivariate analyses with the overall sample and ethnic subgroups, women were consistently found to be more willing to use mental health counseling than men. However, results from other studies have been mixed. Some studies reported that Asian American women had a greater level of willingness to use mental health counseling than Asian American men, while other studies found no gender effect [38–40]. Our finding on gender observed in both the overall sample and ethnic subgroups seems to support the line of literature suggesting that Asian American men face challenges in seeking help for their mental health problems due to cultural expectations and gender norms [41]. Asian American men might be particularly prone to the pressure to be perceived as emotionally strong [41].

It is noteworthy that our hypothesis on mental health needs was not supported. We expected a positive association between mental health need variables and willingness toward mental health counseling, but the outcome was either reversed or non-significant. The presence of mental distress reduced the odds of being willing to use mental health counseling by 17% in the overall sample, and fair/poor ratings of mental health had no impact on willingness. The social stigma toward seeking counseling for mental health problems provides the historical and sociocultural context for our findings [42]. Mental illness and seeking formal services are often perceived as indicative of self-inadequacy and the inability to effectively cope with personal problems, which can be regarded as a source of shame for an individual and their family members [42, 43]. Furthermore, many Asian American communities share similar cultural values and beliefs around collectivism, defining oneself in relation to others within a group or social context, attending to the group's reactions, and consequently managing one's feelings and decisions [42]. Collectivism can also be extended to include indigenous perspectives in which the self is not differentiated from others [44]. Through this worldview, there may be an expectation that individuals make decisions that benefit and please others within a group rather than attending to personal needs [45]. For the Asian American participants in this study, these cultural beliefs and values may have played a role in their low recognition of mental health problems and perceived need for care.

Our hypothesis on the enabling role of prior use of mental health counseling was supported in the overall sample and ethnic subgroups. Given that there have been mixed findings regarding the relationship between prior use of mental health services and future willingness to use counseling, this finding helps to adjudicate the important role that prior use of services has on the positive perceptions and attitudes needed to utilize mental health services among diverse groups of Asian Americans [9, 31–33]. It is noteworthy that the exposure to mental health counseling in the present sample was quite low, less than 10% in the overall sample, and 6.4% to 12.5% in ethnic subgroups had previously used mental health counseling. In our supplementary analyses, those with prior use were more resourced in their English-speaking skills than their counterparts without such an experience. It is speculated that Asian Americans who

are linguistically and culturally adapted are more likely to have access to mental health services and such an exposure leads to a greater acceptance and willingness toward mental health counseling. Given that lack of awareness and stigmatization prohibit the use of mental health counseling, efforts should be made to increase Asian Americans' exposure and openness to mental health services.

Because mental health literacy is a Western framework that uses the biopsychosocial model to understand mental illness, a culturally responsive and adaptive approach that integrates and utilizes traditional ways of understanding and healing would be needed when using this strategy among ethnically and culturally diverse Asian American communities [46]. Mental health researchers, educators, and clinical providers will need to focus their efforts that expand beyond the medical model, to weave social and emotional well-being in collaboration with family members, close friends, community members, and/or spiritual leaders who can help navigate social relationships and communication around mental illness and treatment [4]. This collaboration may facilitate the sharing and integration of cultural knowledge and practice to address mental illness, which may be more helpful in reducing stigma, promoting willingness to receive mental health counseling, and increasing help-seeking [47]. Addressing structural and systemic inequities that impact social determinants of mental health is a critical aspect of this work. The traditional mental health literacy framework may not be applicable to communities facing structural inequities such as immigration status, language barriers, discrimination, and access to culturally appropriate services [48–51]. Modifications would need to be made to incorporate the beliefs, values, and understanding of mental illness and mental health needs, and to acknowledge diverse forms of help that include and encompass alternative interventions that may be more effective in promoting mental health services in ethnically and culturally diverse Asian American communities [47].

This present study poses some limitations. Due to its non-representative and geographically defined sample, the findings may not generalize to the larger population, other regions, or other racial and ethnic groups. Furthermore, we acknowledge the limitation of the AAQoL dataset's cross-sectional study design, from which causal inferences cannot be drawn. Mental health need was also assessed with self-report-based screening tools for non-specific mental distress and self-rated mental health. The use of diagnostic tools for mental health disorders is strongly encouraged. This study also did not include qualitative aspects of prior use (e.g., satisfaction, coping skills learned, perceptions of experiences of mental health counseling, etc.). Furthermore, the high rate of willingness may have been subject to the social desirability of the study participants, which may also help to explain the possible discrepancy between willingness and actual use observed in the target population. Although our findings identified a gap between mental health needs and willingness to use mental health counseling, further attention should be paid to the link between the willingness and actual use of mental health services. Given that the present study examined the general factors regarding mental health and service use using data collected before the COVID-19 pandemic, efforts should be made to revisit the conceptual model in consideration of societal changes and cultural shifts with the advent of the pandemic.

## Conclusion

Despite the limitations, our findings help to illuminate factors associated with the willingness to use mental health counseling among Asian Americans, who are underrepresented in research and underserved in mental health services. Future research and intervention work should focus on strategies to promote mental health literacy and culturally and linguistically appropriate services to increase the use of mental health counseling within Asian American

communities. Moreover, increasing mental health literacy can also promote community support [52]. Applying strategies that are adapted to the diversity of experiences and backgrounds of Asian Americans is imperative. The pandemic and the rise of anti-Asian discrimination illuminates the importance of addressing the presence of barriers to accessing mental health counseling in diverse communities.

## Author Contributions

**Conceptualization:** Ronna Bañada, Yuri Jang.

**Formal analysis:** Ronna Bañada, Yuri Jang.

**Writing – original draft:** Ronna Bañada.

**Writing – review & editing:** Ronna Bañada, Yuri Jang, Lawrence A. Palinkas.

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
