## [Editor Report · Decision Letter 0]

22 Mar 2024

PONE-D-24-06927Willingness to Use Mental Health Counseling in Diverse Groups of Asian AmericansPLOS ONE

Dear Dr. Jang,

Thank you for submitting your manuscript to PLOS ONE. After careful consideration, we feel that it has merit but does not fully meet PLOS ONE’s publication criteria as it currently stands. Therefore, we invite you to submit a revised version of the manuscript that addresses the points raised during the review process.

We have carefully studied and discussed your submission and we would like for the authors to address or highlight the below in the manuscript prior to the review process. Mental health attitudes and services can evolve rapidly, influenced by factors such as societal changes, cultural shifts, and policy developments. Hence, please justify the use of data from 2015: It may or may not be a limitation depending on how stable the factors under study are over time. Kindly justify this.

The AAQoL survey may contain relevant information about the quality of life, cultural attitudes, and potentially attitudes towards mental health services among Asian Americans in central Texas. However, it's crucial to ensure that the survey adequately captures the factors relevant to the willingness to use mental health counseling, post pandemic. Kindly also highlight this in your revision.

We look forward to receiving your revised manuscript.

Kind regards,

Pei Boon Ooi, Ph.D.

Academic Editor

PLOS ONE

Journal Requirements:

"The support for data collection was provided by the City of Austin’s Asian American Quality of Life initiative (Contract No. 26-8275-39, PI−Yuri Jang, Ph.D.).  "

"The support for data collection was provided by the City of Austin’s Asian American Quality of Life initiative (Contract No. 26-8275-39, PI−YJ, Ph.D.)."

"The dataset is publicly available. https://data.austintexas.gov/City-Government/ Final -Repor t-of-the-Asian -Ameri can-Quali ty-of-Life/hc5t-p62z/data"           

4.We note that the grant information you provided in the ‘Funding Information’ and ‘Financial Disclosure’ sections do not match.
---

## [Author Response · Author response to Decision Letter 0]

2 Apr 2024

March 27, 2024

Pei Boon Ooi, Ph.D.

Academic Editor, PLOS One

Dear Dr. Ooi,

Enclosed is our revision of the manuscript entitled “Willingness to Use Mental Health Counseling in Diverse Groups of Asian Americans.” We sincerely appreciate you, as the Academic Editor, and the Reviewers for the thoughtful comments on our work. This letter addresses how we responded to each of the concerns raised. 

Comments

1. Mental health attitudes and services can evolve rapidly, influenced by factors such as societal changes, cultural shifts, and policy developments. Hence, please justify the use of data from 2015: It may or may not be a limitation depending on how stable the factors under study are over time. Kindly justify this.

Response: We sincerely thank the Reviewer for providing feedback to improve the manuscript and have made the necessary revisions.

Given that the present study examined the general factors regarding mental health and service use using data collected before the COVID-19 pandemic, efforts should be made to revisit the conceptual model in consideration of societal changes and cultural shifts with the advent of the pandemic

2. The AAQoL survey may contain relevant information about the quality of life, cultural attitudes, and potentially attitudes towards mental health services among Asian Americans in central Texas. However, it's crucial to ensure that the survey adequately captures the factors relevant to the willingness to use mental health counseling, post pandemic. Kindly also highlight this in your revision.

Response: We thank the Reviewer for this feedback and have revised the manuscript accordingly.

It should be noted that the AAQoL data were collected in central Texas before the COVID-19 pandemic, and the current study addresses the general aspects of mental health service use that may not be specific to location and time. 

Additional Requirements:

Response: We have made the corrections to ensure that the manuscript meets the journal’s style requirements. We added a “Conclusion” section and have made the corrections to the file name.

"The support for data collection was provided by the City of Austin’s Asian American Quality of Life initiative (Contract No. 26-8275-39, PI−Yuri Jang, Ph.D.). "

"The support for data collection was provided by the City of Austin’s Asian American Quality of Life initiative (Contract No. 26-8275-39, PI−YJ, Ph.D.)."

Response: We have made the appropriate corrections and removed funding-related texts from the manuscript.

3. "The dataset is publicly available. https://data.austintexas.gov/City-Government/ Final -Repor t-of-the-Asian -Ameri can-Quality-of-Life/hc5t-p62z/data" 

Response: We thank the Reviewer for bringing our attention to these needed corrections. The funders had no role in study design, data collection and analysis, decision to publish, or preparation of the manuscript.

Response: We have made the necessary corrections in the Funding Information section.

We hope that these revisions improve our manuscript and we hope that you will find it suitable for publication in PLOS One.

Sincerely yours,

---

## [Decision Letter · Decision Letter 1]

21 May 2024

PONE-D-24-06927R1Willingness to Use Mental Health Counseling in Diverse Groups of Asian AmericansPLOS ONE

Dear Dr. Jang,

Thank you for submitting your manuscript to PLOS ONE. After careful consideration, we feel that it has merit but does not fully meet PLOS ONE’s publication criteria as it currently stands. Therefore, we invite you to submit a revised version of the manuscript that addresses the points raised during the review process.

Dear authors:Thank you for your submission. There are minor concerns/suggestions which we invite you to take a look and revise accordingly. ==============================

We look forward to receiving your revised manuscript.

Kind regards,

Pei Boon Ooi, Ph.D.

Academic Editor

PLOS ONE

Journal Requirements:

Reviewers' comments:

Reviewer's Responses to Questions

**Comments to the Author**

1. If the authors have adequately addressed your comments raised in a previous round of review and you feel that this manuscript is now acceptable for publication, you may indicate that here to bypass the “Comments to the Author” section, enter your conflict of interest statement in the “Confidential to Editor” section, and submit your "Accept" recommendation.

Reviewer #1: All comments have been addressed

Reviewer #2: (No Response)

2. Is the manuscript technically sound, and do the data support the conclusions?

Reviewer #1: Yes

Reviewer #2: Yes

3. Has the statistical analysis been performed appropriately and rigorously? 

Reviewer #1: Yes

Reviewer #2: Yes

4. Have the authors made all data underlying the findings in their manuscript fully available?

Reviewer #1: Yes

Reviewer #2: Yes

5. Is the manuscript presented in an intelligible fashion and written in standard English?

Reviewer #1: Yes

Reviewer #2: Yes

6. Review Comments to the Author

Reviewer #1: This paper is good to publish. The outline in this paper is followed the structure of academic journal article. Author is organised well the content of the of study. However, there is a minor correction that need to perform by the author. Detail comments please refer to the attachment file.

Reviewer #2: The study addresses a crucial issue of underutilization of mental health services in Asian American communities and offers valuable insights into the factors influencing their willingness to use mental health counselling. It contributes to the literature on mental health discrepancies and service utilization among Asian Americans.

Introduction:

- Comprehensive introduction and effectively introduces Andersen’s Behavioral Health Model and explains its relevance to understanding individuals’ willingness to use mental health counselling.

- Only a few assertions were made without specific citations, for example, “due to the stigmatization of mental health issues…”, providing citations for the statement would be good.

Method & Results:

- The methodology part is well-structured.

-This study also shows quite a careful consideration of the methodological issue in choosing logistic regression as the primary analytical technique.

-The translation of the survey questionnaire into multiple languages is also a strength of the study.

-The only concern is the use of data from 2015, and whether it is still relevant in capturing contemporary trends – further justification of its use is needed.

Discussion & Conclusion:

-The discussion gives useful inputs, particularly given the underrepresentation of this population in research.

- The study also serves as a foundation for future research and intervention efforts aimed at promoting mental health literacy and culturally competent services within Asian American communities

Overall, I find this manuscript relevant and timely. I recommend this manuscript for publication pending the authors' justification for the use of data from 2015. The clarification regarding the relevance and applicability of the data collected in 2015 is important to ensure the findings remain relevant/appropriate in the current context.

Thank you.

7. PLOS authors have the option to publish the peer review history of their article (what does this mean?). If published, this will include your full peer review and any attached files.

Reviewer #1: **Yes: **AP Dr Nor Mazlina Ghazali

Reviewer #2: No

---

## [Author Response · Author response to Decision Letter 1]

24 May 2024

May 24, 2024

Pei Boon Ooi, Ph.D.

Academic Editor, PLOS One

Dear Dr. Ooi,

Enclosed is our revision of the manuscript entitled “Willingness to Use Mental Health Counseling in Diverse Groups of Asian Americans.” We sincerely thank you and the two Reviewers for the thoughtful comments on our work. This letter addresses how we responded to each of the concerns raised. 

Comments from Reviewer #1

This paper is good to publish. The outline in this paper is followed the structure of academic journal article. Author is organised well the content of the of study. However, there is a minor correction that need to perform by the author. Detail comments please refer to the attachment file.

Response: Thank you for the positive comment on our revision. The remaining issues have been thoroughly addressed. 

Abstract: The researcher is encouraged to insert the implication of study and suggestion to future researcher.

Response: The section has been revised as below.

Given the inverse association between mental health needs and the willingness to use mental health counseling, further attention should be paid to improving Asian Americans’ recognition of mental health symptoms and awareness of the benefit of mental health services. The enabling role of prior use of counseling also highlights the importance of increasing the exposure to mental health services for Asian Americans. In efforts to promote mental health literacy, reduce cultural stigma, and advocate mental health service use, consideration of cultural and linguistic diversity within the Asian American population is imperative. 

Introduction: In Introduction the researcher is encouraged to analyse the information provided in the introduction. It is not only using the information from literature review/previous study.

Response: We would like to note that the introduction section is structured to overview the general literature on the topic and then guide the present investigation. The focus of the current study is on the role of mental health needs and prior use of mental health counseling, and we have structured the introduction section to highlight this uniqueness. Other related variables are included in the assessment as covariates, and we have also addressed their role in the discussion section. 

Research Methodology: 

- Researcher is encouraged to insert the term “Quantitative research design.

- Researcher need to provide the type of sampling technique use in this study.

- Researcher is also need to determine the number of sample size.

- Be specific when you describe the test of data analysis. For instance: descriptive analysis (i.e.: percentage, frequency, mean etc.) and inferential analysis (if related)

- Simplify your table of descriptive analysis

- State AAQoL as instrument in measuring and please state clearly this survey used to measure what variable? 

Response: Each point has been addressed as below. 

- Research design: The following statement has been added: This study utilized a quantitative research design. 

- Sampling technique: The following statement has been added: The AAQoL survey primarily used a convenience sampling approach, and efforts were made to reflect the ethnic composition of the Asian American population in the area. 

- Sample size: The overall sample size of N = 2,609, as well as the sizes of the ethnic subgroups, is noted in the Data section. This information is also added to the heading for Table 1. 

- Test of data analysis: We have specified the test of data analyses, by noting that percentage was reported due to all study variables being in a binary or categorical format. We noted that ethnic group differences were evaluated by using Chi-square analyses. We noted that Spearman’s correlations among the study variables were calculated to understand their underlying associations and ensure the absence of collinearity. We noted that this process was done prior to the multivariate analyses. And noted that a series of logistic regression models were conducted using the overall sample. 

- Simplify your table of descriptive statistics: The readability of the descriptive table has been improved by changing its presentation. We have also added a note to describe the type of analysis. 

- AAQoL instrument: Detailed information on the AAQoL survey along with a citation has been provided. 

This study utilized a quantitative research design. Data came from the Asian American Quality of Life (AAQoL) survey, designed to explore the social and health needs of the growing population of Asian Americans in Central Texas [34]. The survey was conducted with self-identified Asian Americans aged 18 and older between February 1st to December 20th of 2015, in Austin, Texas. 

Discussion: Discussion on result is good, reliable and acceptable. Simplify your table of descriptive analysis.

Response: We are glad to know that our discussion on findings is properly delivered. Please see our response to the comment on the descriptive table above.

Comments from Reviewer #2

The study addresses a crucial issue of underutilization of mental health services in Asian American communities and offers valuable insights into the factors influencing their willingness to use mental health counseling. It contributes to the literature on mental health discrepancies and service utilization among Asian Americans.

Response: The authors sincerely thank Reviewer #2 for their thoughtful input and insights to improve the clarity of the manuscript. We have made edits to address Reviewer #2’s comments.

1. Introduction: 

- Comprehensive introduction and effectively introduces Andersen’s Behavioral Health Model and explains its relevance to understanding individuals’ willingness to use mental health counselling.

- Only a few assertions were made without specific citations, for example, “due to the stigmatization of mental health issues…”, providing citations for the statement would be good.

Response: We appreciate Reviewer #2 for the positive comment on the Introduction section. In this revision, we have added relevant citations. For example, the citation below has been added to the statement on cultural stigma associated with mental health. 

Sanchez F, Gaw A. Mental health care of Filipino Americans. Psychiatric services. 2007 Jun;58(6):810-5.

2. Method and Results:

- The methodology part is well-structured.

- This study also shows quite a careful consideration of the methodological issue in choosing logistic regression as the primary analytical technique. 

- The translation of the survey questionnaire into multiple languages is also a strength of the study.

- The only concern is the use of data from 2015, and whether it is still relevant in capturing contemporary trends – further justification of its use is needed.

Response: While the AAQoL data from 2015 predates the COVID-19 pandemic, many of the variables it addresses, such as health and emotional well-being, are persistent and do not change rapidly. We believe that the methodological rigor and comprehensive nature of the AAQoL survey ensure that its findings remain relevant. These points are addressed in the statements in the methods and discussion sections as below. 

It should be noted that the AAQoL data were collected in central Texas before the COVID-19 pandemic, and the current study addresses the general aspects of mental health service use that may not be specific to location and time.

Given that the present study examined the general factors regarding mental health and service use using data collected before the COVID-19 pandemic, efforts should be made to revisit the conceptual model in consideration of societal changes and cultural shifts with the advent of the pandemic.

3. Discussion & Conclusion: 

- The discussion gives useful input, particularly given the underrepresentation of this population in research.

- The study also serves as a foundation for future research and intervention efforts aimed at promoting mental health literacy and culturally competent services within Asian American communities.

Response: We appreciate Reviewer #2 for the positive comment on the discussion section. 

Overall, I find this manuscript relevant and timely. I recommend this manuscript for publication pending the authors' justification for the use of data from 2015. The clarification regarding the relevance and applicability of the data collected in 2015 is important to ensure the findings remain relevant/appropriate in the current context. 

Response: Please see our response above. 

We hope that these revisions improve our manuscript and we hope that you will find it suitable for publication in PLOS One.

Sincerely yours,

---

## [Editor Report · Decision Letter 2]

7 Jun 2024

PONE-D-24-06927R2Willingness to Use Mental Health Counseling in Diverse Groups of Asian AmericansPLOS ONE

Dear Dr. Jang,

Thank you for submitting your manuscript to PLOS ONE. After careful consideration, we feel that it has merit but does not fully meet PLOS ONE’s publication criteria as it currently stands. Therefore, we invite you to submit a revised version of the manuscript that addresses the points raised during the review process.

Thank you for submitting the revision and we have now hear back from the reviewers. Kindly help to address these comments and we shall then evaluate this submission accordingly.==============================

We look forward to receiving your revised manuscript.

Kind regards,

Pei Boon Ooi, Ph.D.

Academic Editor

PLOS ONE
---

## [Author Response · Author response to Decision Letter 2]

7 Jun 2024

June 7, 2024

Pei Boon Ooi, Ph.D.

Academic Editor, PLOS One

Dear Dr. Ooi,

Enclosed is our revision of the manuscript entitled “Willingness to Use Mental Health Counseling in Diverse Groups of Asian Americans.” Below is a summary of our response to the reviewer’s comment provided in the attachment. 

1. In abstract the researcher is encouraged to insert the implication of study and

suggestion to future researcher. 

Response: We revised the abstract to include the following statements. 

Given the inverse association between mental health needs and the willingness to use mental health counseling, further attention should be paid to improving Asian Americans’ recognition of mental health symptoms and awareness of the benefit of mental health services. The enabling role of prior use of counseling also highlights the importance of increasing the exposure to mental health services for Asian Americans. In efforts to promote mental health literacy, reduce cultural stigma, and advocate mental health service use, consideration of cultural and linguistic diversity within the Asian American population is imperative. 

In Introduction the researcher is encouraged to analyse the information provided in

the introduction. It is not only using the information from literature review/previous study.

Response: We would like to note that the introduction section is structured to overview the general literature on the topic and then guide the present investigation. The focus of the current study is on the role of mental health needs and prior use of mental health counseling, and we have structured the introduction section to highlight this uniqueness. Other related variables are included in the assessment as covariates, and we have also addressed their role in the discussion section. We have also made recommendations for future research. 

Research Methodology

- Researcher is encouraged to insert the term “Quantitative research design.

- Researcher need to provide the type of sampling technique use in this study.

- Researcher is also need to determine the number of sample size.

- Be specific when you describe the test of data analysis. For instance: descriptive

analysis (i.e: percentage, frequency, mean etc) and inferential analysis (if related)

- Simplify your table of descriptive analysis

- State AAQoL as instrument in measuring and please state clearly this survey used to measure what variable? 

Response: Each point has been addressed as below. 

- Research design: The following statement has been added: This study utilized a quantitative research design. 

- Sampling technique: The following statement has been added: The AAQoL survey primarily used a convenience sampling approach, and efforts were made to reflect the ethnic composition of the Asian American population in the area. 

- Sample size: The overall sample size of N = 2,609, as well as the sizes of the ethnic subgroups, is noted in the Data section. This information is also added to the heading for Table 1. 

- Test of data analysis: We have specified the test of data analyses, by noting that percentage was reported due to all study variables being in a binary or categorical format. We noted that ethnic group differences were evaluated by using Chi-square analyses. We noted that Spearman’s correlations among the study variables were calculated to understand their underlying associations and ensure the absence of collinearity. We noted that this process was done prior to the multivariate analyses. And noted that a series of logistic regression models were conducted using the overall sample. 

- Simplify your table of descriptive statistics: The readability of the descriptive table has been improved by changing its presentation. We have also added a note to describe the type of analysis. 

- AAQoL instrument: Detailed information on the AAQoL survey along with a citation has been provided. 

This study utilized a quantitative research design. Data came from the Asian American Quality of Life (AAQoL) survey, designed to explore the social and health needs of the growing population of Asian Americans in Central Texas [34]. The survey was conducted with self-identified Asian Americans aged 18 and older between February 1st to December 20th of 2015, in Austin, Texas. 

Discussion on result is good, reliable and acceptable. Simplify your table of descriptive analysis.

Response: We are glad to know that our discussion on findings is properly delivered. We have also simplified our descriptive table and noted that all values reported in the table are percentages because our study variables are in a binary or categorical format. 

5. Discussion is good and align with result.

6. Conclusion is good.

Response: We appreciate the reviewer for the positive comment on the discussion and conclusion sections. 

We hope that these revisions improve our manuscript and we hope that you will find it suitable for publication in PLOS One.

Sincerely yours,

---

## [Editor Report · Decision Letter 3]

11 Jun 2024

Willingness to Use Mental Health Counseling in Diverse Groups of Asian Americans

PONE-D-24-06927R3

Dear Dr. Jang,

We’re pleased to inform you that your manuscript has been judged scientifically suitable for publication and will be formally accepted for publication once it meets all outstanding technical requirements.

Kind regards,

Pei Boon Ooi, Ph.D.

Academic Editor

PLOS ONE
---

## [Editor Report · Acceptance letter]

20 Jun 2024

PONE-D-24-06927R3 

PLOS ONE

Dear Dr. Jang, 

I'm pleased to inform you that your manuscript has been deemed suitable for publication in PLOS ONE. Congratulations! Your manuscript is now being handed over to our production team.

Kind regards, 

on behalf of

Dr. Pei Boon Ooi 

Academic Editor

PLOS ONE